

# A simple mechanism for unconventional superconductivity in a repulsive fermion model

**Kevin Slagle[1,2⋆] and Yong Baek Kim[1,3]**

**1** Department of Physics, University of Toronto, Toronto, Ontario M5S 1A7, Canada
**2** Department of Physics and Institute for Quantum Information and Matter,
California Institute of Technology, Pasadena, California 91125, USA
**3** Canadian Institute for Advanced Research, Toronto, Ontario, M5G 1M1, Canada

⋆ kslagle@caltech.edu

## Abstract

Motivated by a scarcity of simple and analytically tractable models of superconductivity from strong repulsive interactions, we introduce a simple tight-binding lattice model of fermions with repulsive interactions that exhibits unconventional superconductivity (beyond BCS theory). The model resembles an idealized conductor-dielectric-conductor trilayer. The Cooper pair consists of electrons on opposite sides of the dielectric, which mediates the attraction. In the strong coupling limit, we use degenerate perturbation theory to show that the model reduces to a superconducting hard-core Bose-Hubbard model. Above the superconducting critical temperature, an analog of pseudo-gap physics results where the fermions remain Cooper paired with a large single-particle energy gap.


# 1  Introduction

Understanding unconventional superconductivity [1–9] arising from electron-electron inter-actions is a long-standing problem that has recently been most thoroughly discussed in the context of cuprate [10–16] and iron-based [17–23] superconductors. While there exists a vast literature on this subject, the complexity of these materials is often an obstacle for theoretical modeling. As such, a simple toy model of unconventional (i.e. not phonon-mediated) super-conductivity could be quite useful for strengthening our understanding of electron-interaction-driven superconductivity.

In general, the dominant contributions to an electron Hamiltonian can be modeled on a lattice by a generalized Hubbard model:

$$H = -\sum_{IJ} t_{IJ}\, c_I^\dagger c_J + \sum_{IJ} U_{IJ}\, n_I n_J\,, \tag{1}$$

where $I$ and $J$ include position, orbital, and spin degrees of freedom. The electron hopping $t_{IJ}$ describes the kinetic energy contribution, while the Hubbard interaction $U_{IJ} \geq 0$ describes the repulsive Coulomb force. In order to superconduct, pairs of electrons must form a bosonic bound state, the so-called Cooper pair, which must then condense[1]. It may seem unnatural for electrons to form a bound state due to repulsive interactions. Nevertheless, various mechanisms for how this could occur have been proposed in the literature.

One of the most commonly studied models today is the Hubbard model on a two-dimensional square lattice since it can be thought of as a toy model for the Cuprate superconductors [1]. In the limit of weak Hubbard repulsion, it is possible to analytically show that a Fermi surface instability results in superconductivity in the Hubbard model [25–27]. Although this Fermi surface instability is generic to many models [28–33], it results in an asymptotically small critical temperature [25]. When the repulsion is not weak, a reliable and well-controlled analytical description of the Hubbard model is not known, and numerical studies show evidence for a complicated landscape of many kinds of competing ground states [34]. For large repulsion, the model can be simplified into the t-J model [35], which explicitly contains an attractive interaction. However, a well-controlled analytical description of the t-J model is also not known. Furthermore, there is numerical evidence that the hole-doped Hubbard model does not superconduct in the large repulsion limit ($U \to \infty$) [36].

Hubbard models with spatial inhomogeneity have also been studied [37–39]. In some cases, spatial inhomogeneity can allow for analytical progress resulting in effective hard-core boson models [40, 41]. However, in the large repulsion limit where the analytics are

---

[1]That is, the Hamiltonian must have ground states with $\langle b(x)\rangle \neq 0$, where $b(x) \sim \sum_{IJ}\psi_{IJ}(x)c_I c_J$ is a Cooper pair annihilation operator and $\psi_{IJ}(x)$ is the superconducting order parameter. This occurs when the charge conservation symmetry ($c_I \to e^{i\theta}c_I$) is spontaneously broken, as per Ginzburg-Landau theory. (As is commonly done, we are approximating the electrodynamic gauge field as a classical background field, rather than a dynamical field [24] which would be integrated over in the partition function.)

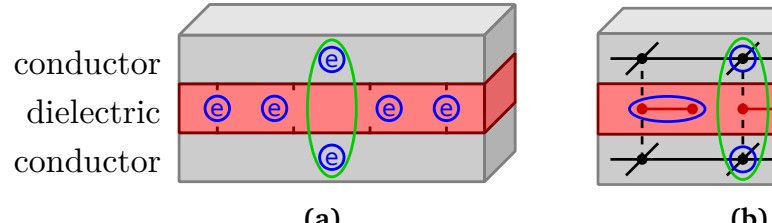

Figure 1: **(a)** A rough cartoon of the model that we study, which resembles an idealized conductor-dielectric-conductor trilayer. An electron on one of the conducting layers repels neighboring electrons on the dielectric, which attracts another electron on the other conducting layer; together, the two electrons form a Cooper pair (circled in green). (See Sec. 5.2 for discussion.) **(b)** The conductor-dielectric-conductor picture can be realized in different ways. In (b), we depict in more detail how it is realized in our toy model [Eq. (6)]. Each blue circle (or oval) denotes the location (or superposition of locations) of an electron for a state with a Cooper pair (circled in green).

simplest, previously-studied models have found either no attraction [41–43] or the strength of the attraction (quantified by the pair-binding energy) approaches zero [41].[2]

Another interesting direction has been to consider electron attraction that results from proximity to a dielectric or semiconductor [44–60]. The idea is similar to phonon-mediated superconductivity, except electrons in a neighboring dielectric or semiconductor play the role that positively charged ions play in BCS superconductivity. However, similar to the Hubbard model, the theoretical analysis of this scenario is also difficult and lacks a well-controlled analytical description.

In this work, we present a simple (and perhaps minimal) tight-binding lattice model of fermions with a strong repulsive interaction that admits a well-controlled analytical description of its superconductivity. Our model resembles an idealized system consisting of a conductor-dielectric-conductor trilayer[3] [see Fig. 1(a)]. For maximal simplicity, we consider spin-polarized electrons (e.g. by a strong in-plane magnetic field) and an anisotropic dielectric. We do not expect that these details are essential for superconductivity, and in Sec. 4.3, we also consider spinful fermions. The Cooper pair has a short coherence-length and consists of electrons on opposite sides of the dielectric, which mediates the attraction. Using degenerate perturbation theory in the limit of a strong repulsive interaction, we show that the model reduces to an s-wave superconducting hard-core boson model.

## 2 Superconductivity Mechanism

A simple mechanism for the emergence of an effective attraction due to repulsive interactions can be understood from the following 4-site spinless fermion model:

---

[2]The model that we introduce is analytically simpler and maintains a finite attraction in the large repulsion limit.

[3]The screening of Coulomb interactions in semiconductor-dielectric-semiconductor trilayers was studied in Ref. [61].

$$
\begin{aligned}
H^{(4)} = &-s\left(c_3^\dagger c_4 + c_4^\dagger c_3\right) \\
&+ U\left(n_1 + n_2\right)n_3 \\
&- \mu\left(n_1 + n_2 + n_3 + n_4\right),
\end{aligned}
\tag{2}
$$

$c_\alpha$ are four spinless fermion annihilation operators with site/orbital index $\alpha = 1,2,3,4$; $n_\alpha = c_\alpha^\dagger c_\alpha$ is the fermion number operator. $s$ is a fermion hopping strength; $U$ is a nearest-neighbor Hubbard repulsion; and $\mu$ is the chemical potential. Sites 3 and 4 could be thought of as a polarizable dielectric, which will mediate an attractive interaction between sites 1 and 2.

It is simplest to consider the limit where $\mu = s/2$ and $s \ll U$. In this limit, the two lowest energy levels of $H^{(4)}$ have the following eigenstates [up to corrections of order $O(s/U)$] and energies [up to $O(s^2/U)$]:

| $\lvert\psi\rangle$ | $E$ |
|---|---|
| $\lvert\tilde{0}\rangle = \frac{1}{\sqrt{2}}\left(c_3^\dagger + c_4^\dagger\right)\lvert0\rangle$ | $-\frac{3}{2}s$ |
| $\lvert\tilde{1}\rangle = c_1^\dagger c_2^\dagger c_4^\dagger\lvert0\rangle$ | |
| $c_1^\dagger c_4^\dagger\lvert0\rangle$ | |
| $c_2^\dagger c_4^\dagger\lvert0\rangle$ | $-s$ |
| $c_3^\dagger c_4^\dagger\lvert0\rangle$ | |
| $c_1^\dagger c_2^\dagger\lvert0\rangle$ | |

$$\tag{3}$$

The ground states are two-fold degenerate, and sites 1 and 2 are either both filled, or neither are filled. As a result, the ground states $\lvert\tilde{0}\rangle$ and $\lvert\tilde{1}\rangle$ act as hard-core boson states with boson number $\eta = 0$ and $\eta = 1$, respectively.

This boson can be thought of as an $s$-wave Cooper pair (correlated with the response of sites 3 and 4), where the fermion antisymmetry exists in the $\alpha = 1,2$ index. If the Cooper pair condenses (in a larger lattice model, e.g. Eq. (6) in the next section), then a superconducting state will result.

If we ignore sites 3 and 4 (e.g. by tracing them out[4]), then the ground states and energy gap of sites 1 and 2 can be roughly described by the following two-site effective Hamiltonian with an attractive interaction

$$
H_{\text{eff}}^{(4)} = -U_{\text{eff}}\left(n_1 - \tfrac{1}{2}\right)\left(n_2 - \tfrac{1}{2}\right),
\tag{4}
$$

where $\frac{1}{2}U_{\text{eff}} = \frac{1}{2}s + O(s^2/U)$ is the energy-gap to the excited states in Eq. (3). The attractive interaction can also be quantified by a positive pair-binding energy

$$
\Delta_{\text{pb}} = 2E_2 - (E_1 + E_3) = s \quad \text{when } s \ll U,
\tag{5}
$$

where $E_n$ is the lowest possible energy of a state with $n$ fermions. The physics is similar for smaller $U/s$ (see Fig. 2), but the analytical expressions are slightly more complicated [see e.g. Eq. (27) in the appendix].

To understand the effective attractive interaction, note that if there is a fermion at site 1, then the strong repulsion $U$ will prevent the occupation of site 3. Due to the negative chemical

---

[4]If we take the low-temperature ($\beta s \gg 1$) density matrix $\rho = \frac{1}{Z}e^{-\beta H^{(4)}}$ (where $Z = \operatorname{tr} e^{-\beta H^{(4)}}$) of $H^{(4)}$ [Eq. (2)], and trace out sites 3 and 4, then the resulting density matrix is equal to the density matrix $\rho_{\text{eff}}$ of $H_{\text{eff}}^{(4)}$ (also known as the entanglement Hamiltonian [62, 63]) up to $O(e^{-\frac{3}{2}\beta s})$ corrections. That is, $\rho_{\text{eff}} = \operatorname{tr}_{34}\rho = \sum_{n_1',n_2'=0,1}^{n_1,n_2,n_3,n_4} \lvert n_1 n_2\rangle\langle n_1 n_2 n_3 n_4\rvert\rho\lvert n_1' n_2' n_3 n_4\rangle\langle n_1' n_2'\rvert + O(e^{-\frac{3}{2}\beta s})$

ground state electron number

pair-binding energy

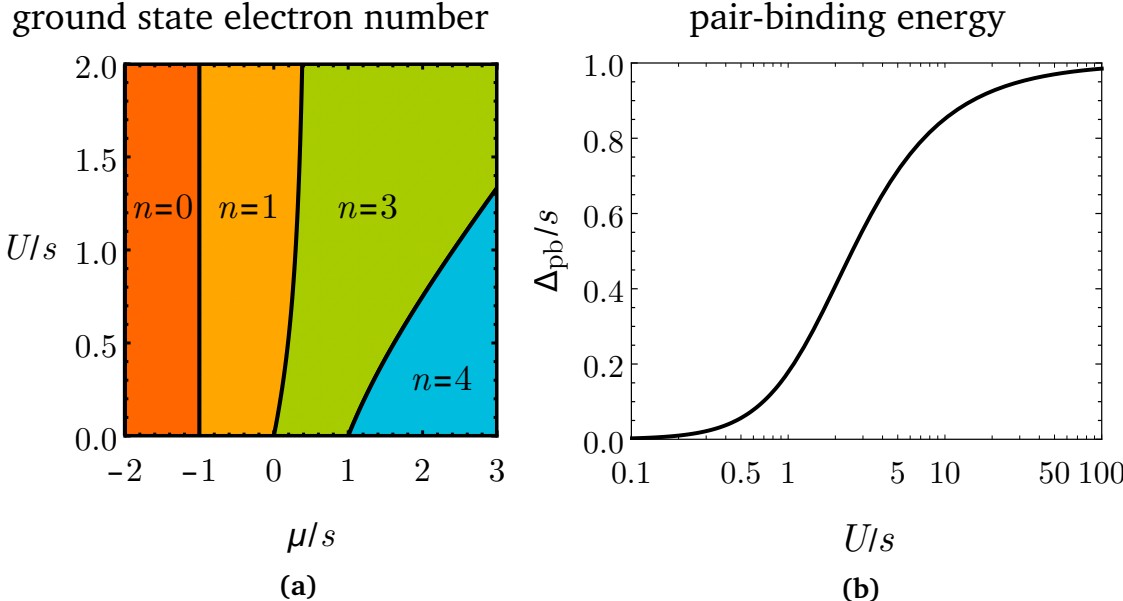

Figure 2: **(a)** The number of electrons in the ground state of the 4-site model [Eq. (2)] as a function of the chemical potential $\mu$ and Hubbard repulsion $U$ (relative to the hopping strength $s$). Cooper pairing exists near the boundary between the $n = 1$ and $n = 3$ region since this is where there are degenerate ground states with a difference in fermion number equal to two. **(b)** Pair-binding energy [Eq. (5)] as a function of the interaction strength $U/s$. The effective attractive interaction becomes stronger as the strength of the Hubbard interaction increases.

potential, the ground state ($c_1^\dagger c_2^\dagger c_4^\dagger |0\rangle$) prefers to fill sites 2 and 4. Alternatively, if sites 1 and 2 are empty, then a single fermion can resonate between sites 3 and 4, leading to the state $\frac{1}{\sqrt{2}}\left(c_3^\dagger + c_4^\dagger\right)|0\rangle$. We have tuned $\mu/s$ such that these two cases result in equal-energy ground states. Thus, at low energy, sites 1 and 2 are either both filled or are both empty, which is in accordance with the effective attraction in Eq. (4).

Thus, $H^{(4)}$ [Eq. (2)] demonstrates a simple mechanism for how a fermion hopping model with strong repulsive interactions can lead to an effective attractive interaction.

## 3 Minimal Model

By using $H^{(4)}$ to generate an effective attractive interaction, we can write down a simple repulsive fermion model with a superconducting ground state in the limit of strong interactions. The model is simply a grid of coupled $H^{(4)}$ models:

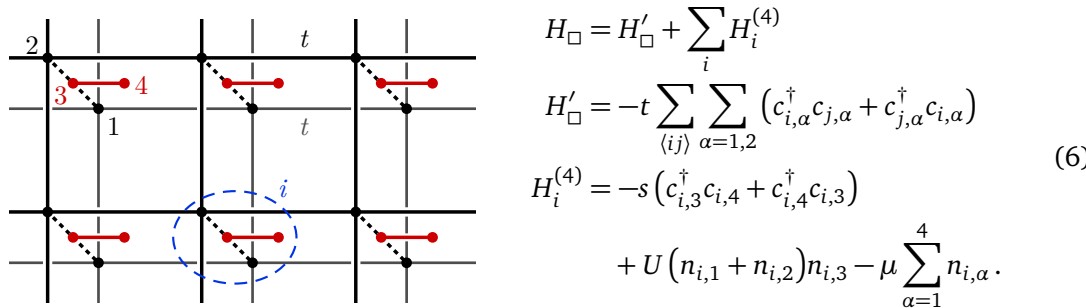

$$H_\square = H'_\square + \sum_i H_i^{(4)}$$

$$H'_\square = -t \sum_{\langle ij \rangle} \sum_{\alpha=1,2} \left(c_{i,\alpha}^\dagger c_{j,\alpha} + c_{j,\alpha}^\dagger c_{i,\alpha}\right)$$

$$H_i^{(4)} = -s\left(c_{i,3}^\dagger c_{i,4} + c_{i,4}^\dagger c_{i,3}\right)$$

$$+ U\left(n_{i,1} + n_{i,2}\right)n_{i,3} - \mu \sum_{\alpha=1}^{4} n_{i,\alpha}.$$

(6)

Here, we are considering two two-dimensional square-lattice layers (black) which interact via the intermediate red sites. The red sites resemble an idealized (and anisotropic) dielectric in a spin-polarized conductor-dielectric-conductor trilayer [see Fig. 1(b)] since the red layer is insulating and polarizable. $\sum_{\langle ij \rangle}$ sums over all pairs of nearest-neighbor unit cells $i$ and $j$. (A unit cell is circled in blue above.) Each unit cell $i$ is composed of an $H^{(4)}$ model, which includes four spin-less fermions $c_{i,\alpha}$ indexed by $\alpha = 1, 2, 3, 4$. $H'_\square$ adds a hopping term $t$ for the $\alpha = 1, 2$ fermions. We will focus on the following limit[5]:

$$|\mu - s/2| < t \ll s \ll U. \tag{7}$$

In this limit, each $H_i^{(4)}$ will approximately always be in one of its two ground states [Eq. (3)]. The fermion hopping term in $H'_\square$ couples the $H_i^{(4)}$ models together. But in order for each $H_i^{(4)}$ to remain in its ground state, the perturbation $H'_\square$ must act twice in order to move two fermions (a Cooper pair) from one site to another. Thus, we can use degenerate perturbation theory [64–66] to obtain a low-energy effective Hamiltonian. (See Appendix B for details.) The resulting model can be written in the form of the following hard-core boson model:

$$H_\square^{\text{eff}} = -t_{\text{eff}} \sum_{\langle ij \rangle} \left( b_i^\dagger b_j + b_j^\dagger b_i \right) - 2\mu' \sum_i \eta_i + V_{\text{eff}} \sum_{\langle ij \rangle} \left( \eta_i - \tfrac{1}{2} \right) \left( \eta_j - \tfrac{1}{2} \right), \tag{8}$$

$$V_{\text{eff}} = 2 t_{\text{eff}}, \qquad t_{\text{eff}} = t^2/s, \qquad \mu' = \mu - \frac{1-\epsilon}{2} s, \tag{9}$$

where $\epsilon = \sqrt{1 + (U/s)^2} - U/s = \frac{s}{2U} + O(s/U)^3$. $\mu'$ is defined such that the 4-site model is exactly degenerate when $\mu' = 0$. The hard-core constraint implies that the boson number operator $\eta_i = b_i^\dagger b_i = 0, 1$. The above effective Hamiltonian contains all terms that are not smaller than $O(t^2/s)$. ($H_\square^{\text{eff}}$ can also be transformed into an XXZ spin model in a magnetic field[6].)

Physically, the boson is a Cooper pair of the fermions: $b_i \sim c_{i,1} c_{i,2}$. The boson hopping term $t_{\text{eff}} b_i^\dagger b_j$ results from a virtual process $(t \, c_{i,2}^\dagger c_{j,2})(t \, c_{i,1}^\dagger c_{j,1})$ that hops two fermions from site $j$ to $i$. The nearest-neighbor repulsion $V_{\text{eff}} (\eta_i - \tfrac{1}{2})(\eta_j - \tfrac{1}{2})$ results from a virtual process $(t \, c_{j,\alpha}^\dagger c_{i,\alpha})(t \, c_{i,\alpha}^\dagger c_{j,\alpha})$ where a fermion hops from site $j$ to $i$ and then back to $j$. In both virtual processes, the intermediate state has a large energy $s \gg t$, which penalizes the virtual processes and results in the energy scaling $t^2/s$ for $t_{\text{eff}}$ and $V_{\text{eff}}$.

The phase diagram of this effective boson model is shown in Fig. 3. The ground state is in a superfluid phase when $0 \neq |\mu'| < 4t_{\text{eff}}$ (and $V_{\text{eff}} = 2t_{\text{eff}}$). Since the effective boson carries the same charge as two fermions, the superfluid in the effective model corresponds to a superconductor in the original fermion model $H_\square$ [Eq. (6)]. Therefore, the ground state of $H_\square$ is a superconductor in the limit of interest [Eq. (7)] when $0 \neq |\mu'| < 4t^2/s$.

One may worry that the effective boson model requires fine tuning. However, this is not the case; the 4-site model [Eq. (2)] was fine-tune to be degenerate only so that we could conveniently apply degenerate perturbation theory. Any sufficiently-small local perturbation can be added to the fermion model without destroying the superconductivity. The only result is that the coefficients in the effective boson model are shifted, or new terms could be generated.

---

[5]The $s \ll U$ assumption is not necessary for analytical tractability and obtaining a superconducting hard-core boson model, but it makes the analysis much simpler. Arbitrary $U > 0$ can be considered by working harder in Eq. (37) of the appendix. (However, if $U \ll s$, then $t$ will have to be much smaller than the single-fermion excitation gap, $t \ll \Delta_{\text{pb}} \sim \frac{1}{4} U^2/s$ [Fig. 2(b)], in order for the perturbative methods used in Appendix B.1 to be applicable.)

[6]$H_\square^{\text{eff}}$ in Eq. (8) can be viewed as an XXZ spin-1/2 model $H_\square^{\text{eff}} = -\frac{1}{2} t_{\text{eff}} \sum_{\langle ij \rangle} (\sigma_i^x \sigma_i^x + \sigma_i^y \sigma_i^y) + \frac{1}{4} V_{\text{eff}} \sum_{\langle ij \rangle} \sigma_i^z \sigma_i^z -$

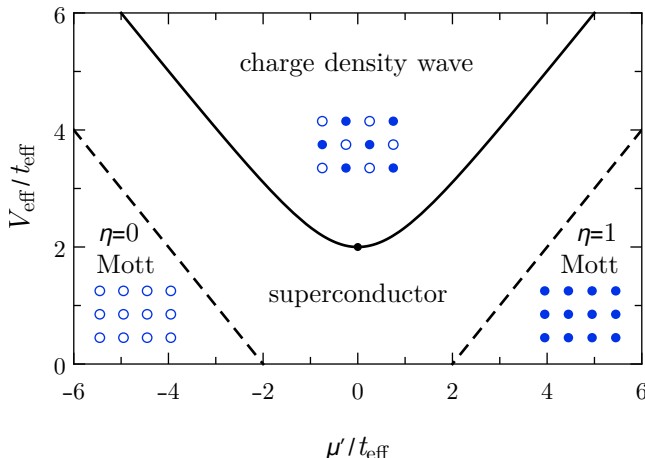

Figure 3: A phase diagram of the hard-core boson model [Eq. (8)] extracted from Ref. [67]. A large boson chemical potential $\mu'$ results in a Mott insulating phase with boson number $\eta = 0$ or $\eta = 1$ on every site. A large boson repulsion $V_{\text{eff}}$ induces a charge density wave where half of the hard-core boson states are filled in a checkerboard pattern. In between these phases is a superconducting phase. The fermion model [Eq. (6)] results in $V_{\text{eff}}/t_{\text{eff}} = 2$. The phase transitions across the dashed lines are continuous, while the transition across the solid line is discontinuous [68]. (There is a hidden $SU(2)$ symmetry when $\mu' = 0$ and $V_{\text{eff}}/t_{\text{eff}} = 2$.)

## 4 Extensions

Because theoretical simplicity was the primary goal for our model [Eq. (6)], it is natural that some aspects of it are not realistic. In this section, we will exemplify possible ways that the model could be extended to make it more realistic. We show that for each of these extensions, the model still reduces to a hard-core boson model that is either known to or is very likely to exhibit superconductivity. In Appendix A, we will also explore an alternative lattice geometry. An actual material may realize more than one of these extensions.

### 4.1 Missing Hopping

The absence of a fermion hopping between sites 3 and 1 (and between 3 and 2) may seem peculiar. Let us then consider the effect of extending the 4-site model [Eq. (2)] with such a hopping:

$$H_{t'} = -t'(c_1^\dagger c_3 + c_3^\dagger c_1) - t'(c_2^\dagger c_3 + c_3^\dagger c_2). \tag{10}$$

If the hopping energy $t'$ is much less than the energy gap $s/2$, then the low-energy eigenstates and energies in Eq. (3) will not change. But if $t' \gtrsim s/2$, then the low-energy states and energy spectrum will change significantly, which will likely destroy the superconducting ground state when the 4-site clusters are coupled together.

However, we should also consider the effect of a repulsive interaction between sites 3 and 4, which we will write as:

$$H_{U'} = U'(n_3 + n_4 - 1)^2. \tag{11}$$

---

$\mu' \sum_i \sigma_i^z$ by replacing the hard-core boson operator $b_j \rightarrow \frac{1}{2}(\sigma_j^x + i\sigma_j^y)$ with Pauli operators $\sigma_i^\mu$. Since $V_{\text{eff}} = 2t_{\text{eff}}$, $H_\square^{\text{eff}}$ can also be transformed into an $SU(2)$ anti-ferromagnetic Heisenberg model in an applied field $H_\square^{\text{eff}} = \frac{1}{2} t_{\text{eff}} \sum_{\langle ij \rangle} \vec{\sigma}_i \cdot \vec{\sigma}_j - \mu' \sum_i \sigma_i^z$ by rotating the spins on the A sublattice by the unitary operator $U = \prod_{i \in A} \sigma_i^z$.

Since $U$ is large, it is natural to also consider the possibility that $U'$ is also large. In this case, $U'$ energetically forbids states that do not have a total of one fermion on sites 3 and 4. Therefore, if we consider extending the 4-site model by these terms, $H^{(4)} + H_{t'} + H_{U'}$ [Eqs. (2), (10), (11)], then this extended model will have the same low-energy eigenstates and energies in Eq. (3) as the original 4-site model as long as $t' \ll \max(s, U')$ (and $\mu = s/2$ and $s \ll U$ as before). Therefore, when the 4-site clusters are coupled together, the additional hopping $t'$ will not hamper superconductivity as long as $t' \ll \max(s, U')$.

## 4.2 Covalent Bonds

If we were to look for a crystal or molecule realization of the 4-site cluster [Eq. (2)], then the geometry of the 4-site cluster [Eq. (2)] may seem unnatural due to site 4, which only couples to a single site. However, the cluster can easily be expanded. For example, the middle section could be modified to model a covalent bond between two or more ions (or nuclei).

We will now discuss the example involving a covalent bond between two ions, which can be modeled by the following 5-site cluster:

$$
\begin{aligned}
H^{(5)} = &-s\left(c_3^\dagger c_4 + c_4^\dagger c_3\right) \\
&-s\left(c_3^\dagger c_5 + c_5^\dagger c_3\right) \\
&+ U\left(n_1 + n_2\right)n_3 \\
&-\mu\left(n_1 + n_2 + n_3 + n_4 + n_5\right),
\end{aligned}
\tag{12}
$$

$c_\alpha$ are five spinless fermion annihilation operators with site/orbital index $\alpha = 1, 2, 3, 4, 5$. Sites 3, 4, and 5 model a covalent bond between a pair of ions at sites 4 and 5. Site 3 is (very coarsely) modeling the electron states between the two ions. One could think of sites 4 and 5 as $s$ orbitals, while site 3 can be thought of as the superposition of $p_x$ orbitals of ions 4 and 5 that constructively interferes in the area between the ions. Other combinations of orbitals are also possible.

When $\mu = s/\sqrt{2}$ and $s \ll U$, the two lowest energy levels are:

| $|\psi\rangle$ | $E$ |
|---|---|
| $|\tilde{0}\rangle = \frac{1}{2\sqrt{2}}(c_4^\dagger + \sqrt{2}c_3^\dagger + c_5^\dagger)(c_4^\dagger - c_5^\dagger)|0\rangle$ | $-2\sqrt{2}s$ |
| $|\tilde{1}\rangle = c_1^\dagger c_2^\dagger c_4^\dagger c_5^\dagger |0\rangle$ | |
| 6 degenerate states | $-\frac{3}{2}\sqrt{2}s$ |

(13)

Similar to the 4-site model [Eq. (3)], the ground states are two-fold degenerate and act as hard-core boson states. When an electron is at site 1 or 2, the covalent bond is damaged since the repulsive interaction $U$ prevents fermions from hopping onto site 3. In the covalent bond picture, the covalent bond mediates an effective attractive interaction [of the form of Eq. (4)] between the fermions on sites 1 and 2, and a filled hard-boson corresponds to a damaged covalent bond.

If many 5-site clusters are weakly coupled together in a grid [similar to Eq. (6)], then the low-energy physics can be effectively described by a hard-core boson model [Eq. (8) with $V_{\text{eff}} = 2t_{\text{eff}}$ and $t_{\text{eff}} = t^2/\sqrt{2}s$]. A superconducting ground state results when $0 \neq |\mu'| \lesssim 4t_{\text{eff}} = 2\sqrt{2}t^2/s$, where $\mu' = \mu - s/\sqrt{2} + O(s^2/U)$.

## 4.3 Spinful Fermions

The previous models have all involved spinless fermions. But electrons are spin-half particles. An applied in-plane magnetic field could gap out the spin degree of freedom and effectively

result in the spinless fermion model in Sec. 3. However, in this section, we will show that if a spin degree of freedom is added to the fermions in the 4-site model, then a superconducting ground state may still result as long as a large on-site Hubbard repulsion is included.

A spinful generalization of the 4-site model is:

$$
H^{\text{spin}} = -s \sum_{\sigma=\uparrow,\downarrow} \left( c_{3,\sigma}^\dagger c_{4,\sigma} + c_{4,\sigma}^\dagger c_{3,\sigma} \right)
$$
$$
+ U_0 \sum_{\alpha=1,2} n_\alpha (n_\alpha - 1) \tag{14}
$$
$$
+ U_1 (n_1 + n_2) n_3
$$
$$
- \mu_{12} (n_1 + n_2) - \mu_{34} (n_3 + n_4),
$$

where $n_\alpha = \sum_\sigma n_{\alpha,\sigma}$ is the total fermion number on site $\alpha$ and $\sigma = \uparrow, \downarrow$ denotes the two electron spin states.

When $\mu_{12} = 2\mu_{34} = s$, $s \ll U_0$, and $s \ll U_1$, the lowest energy levels are:

$$
\begin{array}{c|c}
|\psi\rangle & E \\
\hline
|\tilde{0}\rangle = \frac{1}{2} \left( c_{3\uparrow}^\dagger + c_{4\uparrow}^\dagger \right) \left( c_{3\downarrow}^\dagger + c_{4\downarrow}^\dagger \right) |0\rangle & -3s \\
|\tilde{1}_{\sigma\sigma'}\rangle = c_{1,\sigma}^\dagger c_{2,\sigma'}^\dagger c_{4\uparrow}^\dagger c_{4\downarrow}^\dagger |0\rangle & \\
\hline
\text{10 degenerate states} & -\frac{5}{2}s
\end{array} \tag{15}
$$

The ground states are five-fold degenerate. $|\tilde{1}_{\sigma\sigma'}\rangle$ denotes four different spin states indexed by $\sigma, \sigma' = \uparrow, \downarrow$. Therefore, the low-energy states behave like a hard-core boson with four different spin states.

If many spinful 4-site clusters are coupled together in a grid [similar to Eq. (6)], then the low-energy physics can be effectively described by a hard-core boson model [similar to Eq. (8)] where the boson has four spin states. It is very likely that this hard-core boson model has a superconducting ground state in some regions of its phase diagram. However, additional perturbations should be added to the model since they will generically split the degeneracy between the spin singlet and triplet states of the hard-core boson.

### 4.3.1 Spin-singlet Case

The simplest perturbation to consider is the following hopping between sites 1 and 2:

$$
H_{t''} = -t'' \left( c_{1,\sigma}^\dagger c_{2,\sigma} + c_{2,\sigma}^\dagger c_{1,\sigma} \right). \tag{16}
$$

This perturbation splits the 4-fold degeneracy of the 4-fermion states $|\tilde{1}_{\sigma\sigma'}\rangle$ such that the spin-singlet state $\frac{1}{\sqrt{2}} \left( |\tilde{1}_{\uparrow\downarrow}\rangle - |\tilde{1}_{\downarrow\uparrow}\rangle \right)$ is preferred by an energy splitting equal to $(t'')^2/U$ (at leading order).[7]

When the spinful 4-site clusters are coupled together by a hopping $t$, similar to Eq. (6), then the same procedure used in Sec. 3 can be used to derive the exact same superconducting hard-core boson model in Eq. (8), but with slightly different coefficients from those in Eq. (9). This hard-core boson model is valid as long as $t \ll (t'')^2/U$; if this is not the case, one should instead consider a hard-core boson model with the four different spin states for each boson (corresponding to the four $|\tilde{1}_{\sigma\sigma'}\rangle$ states).

---

[7]One could also consider a nearest-neighbor hopping from sites 1 and 2 to 3: $t'(c_{1,\sigma}^\dagger + c_{2,\sigma}^\dagger)c_{3,\sigma} + h.c.$, similar to Eq. (10). Such a term would also favor the spin-singlet state, but it would only split the 4-fold degeneracy of the 4-fermion ground states by $O((t')^4/sU^2)$.

# 5    Discussion

We have considered a simple two-dimensional lattice model [Eq. (6)] with a superconducting ground state. The primary motivation of our work was to uncover the simplest possible analytically-tractable model of superconductivity in the strong repulsion limit. However, the model also has a number of other interesting features and possible applications, which we will discuss.

The Cooper pairing is ultimately a result of the local Coulomb repulsion physics in the 4-site fermion model [Eq. (2)], and the size of the Cooper pair is just a single unit cell. Because the Cooper pairing results from charge interactions, it is interesting to note that the superconducting phase neighbors a charge density wave order (Fig. 3), rather than an antiferromagnetic order as in the cuprate materials.

## 5.1    Finite Temperature

At temperatures below the single-particle fermion gap $s/2$ [from Eq. (3)], the model is well-approximated by a hard-core boson model [Eq. (8)] with hopping strength $t_{\text{eff}} \sim t^2/s$. If we consider a 3D stack of the 2D model with a weak fermion hopping between the stacks, then the resulting three-dimensional model can be expected[8] to exhibit superconductivity at temperatures below $T_c \sim t_{\text{eff}}/2 \sim t^2/2s$ [67]. Although we only considered a single corner [Eq. (7)] of the phase diagram, any sufficiently-small local perturbation can be added without destroying the superconductivity.

At temperatures above the superconducting critical temperature $T_c$ but below $T^\star \sim s$, the fermions are Cooper paired with a gap $\Delta \approx s/2$ to single-fermion excitations, which realizes an effective hard-core boson model. Ref. [70] showed that the DC (zero frequency) resistivity of the hard-core boson model [Eq. (8)] with $\mu' = V_{\text{eff}} = 0$ is[9]

$$\rho \approx 0.23 \frac{h}{4e^2} k_{\text{B}} T/t_{\text{eff}} \tag{17}$$

at high temperatures, which could apply to our fermion model $H_\square$ [Eq. (6)] in the temperature range $t_{\text{eff}} \ll T \ll s$. This regime of a large single-particle gap and large resistivity (linear in temperature) therefore appears to be an analog of the pseudo-gap physics [71–74] seen in the cuprate and iron-based superconductors. However, Ref. [70] only considered a hard-core boson model at half filling and without a nearest-neighbor Hubbard repulsion ($\mu' = V_{\text{eff}} = 0$). Future work is required to determine the robustness of the large linear resistivity ($\rho \propto T$) to these perturbations, which are present in our low-energy boson models.

It could also be interesting to investigate the physics above temperature $T^\star \sim s$. One possibility is a crossover to strange/bad metal physics [75–84], i.e. a large resistivity linear in temperature without a large single-particle gap.

## 5.2    Possible Physical Realizations

Let us speculate on possible physical realizations of our model. As discussed in Sec. 4, the model can be extended in various ways that could help facilitate a material realization. For example, the effective spinless fermion model [Eq. (6)] could result from an applied (or

---

[8]Before coupling the stacks, each layer is in a state with quasi-long-range superconducting order below a Berezinskii-Kosterlitz-Thouless transition [69] critical temperature $T_{\text{KT}} \sim t_{\text{eff}}/2$ [67]. Since the correlation length of each layer is infinite, a weak fermion coupling between the layers will result in a long-range superconducting order with roughly the same critical temperature $T_c \sim T_{\text{KT}}$.

[9]This resistivity is obtained from Eq. (75) of Ref. [70]. $k_{\text{B}}$ is Boltzmann's constant, and $h/4e^2$ is the quantum of resistance for a charge $q = 2e$ Cooper pair, where $h$ is Planck's constant.

induced) in-plane magnetic field, which can gap out the spin degree of freedom. Such an example is of practical interest since it can result in a superconductor that is more robust to strong magnetic fields.

An interesting possible realization would be a conductor-dielectric-conductor trilayer. To see the connection to our model: note that in Eq. (6), (14), or (20), the middle layer of red sites resembles an idealized (and anisotropic) dielectric since it is highly polarizable and insulating; and the middle layer is neighbored by conducting layers. Taking inspiration from the large superconducting $T_c$ at the interface of $FeSe/SrTiO_3$ [54, 55] or $FeSe/TiO_2$ [56], one of many possible material candidates could be a $FeSe$-$TiO_2$-$FeSe$ trilayer where a single dielectric $TiO_2$ layer neighbors single layers of conducting $FeSe$. Considering a conductor-dielectric-conductor trilayer, rather than a single conductor/dielectric interface [52–56], may help increase $T_c$ since the Cooper-paired electrons are separated further apart, which decreases the repulsive Coulomb interaction that the emergent attraction must overcome. However, our model is only a toy model for such a situation and more detailed future study is warranted.

Another possibility is to think of the 4-site model [Eq. (2)] as a minimal model for a molecule, similar to Ref. [85].[10] If a molecule with similar physics can be discovered, then a liquid or crystal of such molecules could exhibit superconductivity. In particular, the lowest-energy states of the molecule should have fermion occupation numbers that differ by two, as in Eq. (3). In fact, this kind of physics has already been shown to occur in doped buckminsterfullerene $C_{60}$ molecules [86–88].

One could also view the 4-site model as a simplified toy model for a recent carbon nanotube experiment [58].

# Acknowledgements

We thank Garnet Chan, Alon Ron, Arun Paramekanti, Patrick Lee, Alex Thomson, Jong Yeon Lee, Nai-Chang Yeh, Yuval Oreg, Arbel Haim, Leonid Isaev, and Olexei Motrunich for helpful discussions.

**Funding information**  This work was supported by the NSERC of Canada and the Center for Quantum Materials at the University of Toronto. KS also acknowledges support from the Walter Burke Institute for Theoretical Physics at Caltech.

# A  Bilayer Triangular Lattice

In this appendix, we will exemplify another nontrivial way in which the minimal model in Sec. 3 can be modified so as to have a lattice structure that is more likely to have a material realization.

---

[10]Our model can be thought of a simplified version of the model in Ref. [85]. Their $d_{xz}$ and $p_1$ (and $d_{yz}$ and $p_2$) orbitals are similar to sites 3 and 4 [red in Eq. (2)] in the 4-site model. However, the geometry of their model is less favorable for Cooper pairing since sites 1 and 2 in our model become two different spins of a $d_{zz}$ orbital in their model; this is because the double occupancy of a $d_{zz}$ orbital has a large Coulomb energy cost, which weakens Cooper pairing and is why their $U_{eff}$ is always non-negative in their Fig 2. To overcome this, Ref. [85] considers a larger and more complicated model to obtain an effective attraction.

We will consider the following 6-site spinless fermion model:

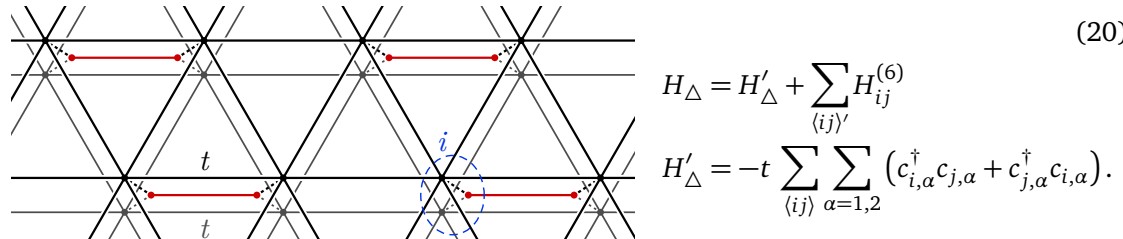

$$
\begin{aligned}
H_{ij}^{(6)} = & -s\left(c_{i,3}^{\dagger}c_{j,3} + c_{j,3}^{\dagger}c_{i,3}\right) \\
& + U\sum_{i'=i,j}\left(n_{i',1} + n_{i',2}\right)n_{i',3} \\
& + V\left(n_{i,1}n_{j,1} + n_{i,2}n_{j,2}\right) \\
& - \mu\sum_{i'=i,j}\left(n_{i',1} + n_{i',2} + n_{i',3}\right),
\end{aligned}
\tag{18}
$$

$i$ and $j$ index different 3-site clusters, while $\alpha = 1, 2, 3$ indexes the three sites within a cluster. $H_{ij}^{(6)}$ couples two 3-site clusters ($i$ and $j$) together.

When $V = \mu = s/2$ and $s \ll U$, the two lowest energy levels of $H_{ij}^{(6)}$ are:

| $\lvert\psi\rangle$ | $E$ |
|---|---|
| $\lvert\widetilde{00}\rangle = \frac{1}{\sqrt{2}}\left(c_{i,3}^{\dagger} + c_{j,3}^{\dagger}\right)\lvert0\rangle$ | |
| $\lvert\widetilde{10}\rangle = c_{i,1}^{\dagger}c_{i,2}^{\dagger}c_{j,3}^{\dagger}\lvert0\rangle$ | $-\frac{3}{2}s$ |
| $\lvert\widetilde{01}\rangle = c_{j,1}^{\dagger}c_{j,2}^{\dagger}c_{i,3}^{\dagger}\lvert0\rangle$ | |
| 14 degenerate states | $-s$ |

$$\tag{19}$$

The lowest energy level is now triply degenerate. But since the three low-energy states each differ by an even number of fermions, we can still think of them as hard-core boson states $\lvert\widetilde{\eta_i\eta_j}\rangle$ with fillings $\eta_i, \eta_j = 0, 1$ but where the $\lvert\widetilde{11}\rangle$ state is gapped out due to a large effective bosonic repulsive interaction. If this effective boson condenses, then a superconducting state will result.

To achieve this, we will embed $H_{ij}^{(6)}$ into a layered triangular lattice:

$$
\begin{aligned}
H_{\triangle} &= H_{\triangle}' + \sum_{\langle ij\rangle'}H_{ij}^{(6)} \\
H_{\triangle}' &= -t\sum_{\langle ij\rangle}\sum_{\alpha=1,2}\left(c_{i,\alpha}^{\dagger}c_{j,\alpha} + c_{j,\alpha}^{\dagger}c_{i,\alpha}\right).
\end{aligned}
\tag{20}
$$

Similar to Eq. (18), $i$ and $j$ index the different 3-site clusters, which are located at the vertices of a triangular lattice. $\sum_{\langle ij\rangle}$ sums over all nearest-neighbor 3-site clusters (along the solid gray and black lines), while $\sum_{\langle ij\rangle'}$ only sums over the neighboring 3-site clusters with a red line between them. We will focus on the following limit:

$$
|\mu - s/2| < t \ll V = s/2 \ll U.
\tag{21}
$$

Again, we can use degenerate perturbation theory to derive a low-energy effective hard-

core boson model (see Appendix B.2 for details):

$$H_\triangle^{\text{eff}} = -\frac{t^2}{s} \sum_{\langle ij \rangle''} \left( b_i^\dagger b_j + b_j^\dagger b_i \right) - 2\mu' \sum_i \eta_i$$
$$+ \frac{2t^2}{s} \sum_{\langle ij \rangle''} (1 - \eta_{\hat{i}}) \underbrace{(\eta_i \eta_j - \tfrac{1}{2}\eta_i - \tfrac{1}{2}\eta_j)}_{\left( \eta_i - \frac{1}{2} \right)\left( \eta_j - \frac{1}{2} \right) - \frac{1}{4}} (1 - \eta_{\hat{j}}) \tag{22}$$

$$\mu' = \mu - \frac{s}{2} + O\left( \frac{s^2}{U} \right) \tag{23}$$

constraint: $\eta_i \eta_{\hat{i}} = 0$ across dashed red links

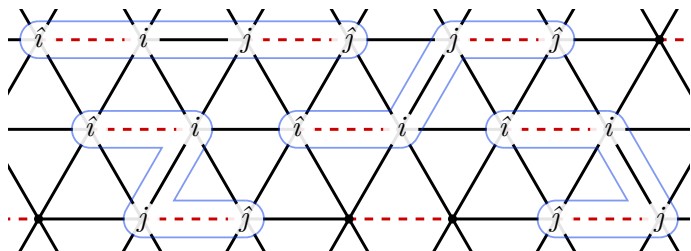

$\sum_{\langle ij \rangle''}$ sums over neighboring sites across a black link. The last term sums over every pair of neighboring sites $\langle ij \rangle$ across a black link; we then define the $\hat{i}$ in $(1 - \eta_{\hat{i}})$ to be the site across the dashed red link from $i$, and similar for $\hat{j}$ and $j$. Above, we highlight four examples in blue of the $(\hat{i}, i, j, \hat{j})$ that are summed over. For each $\langle i\hat{i} \rangle$ across a dashed red link, a $\eta_i \eta_{\hat{i}} = 0$ constraint results because none of the three low-energy states [Eq. (19)] across a dashed red link correspond to a state with two bosons. The hard-core boson number operator is $\eta_i = b_i^\dagger b_i = 0, 1$. Physically, the boson is a Cooper pair of the fermions: $b_i \sim c_{i,1} c_{i,2}$.

The last term in $H_\triangle^{\text{eff}}$ is a four-boson repulsion term that results from a virtual process $(t\, c_{j,\alpha}^\dagger c_{i,\alpha})(t\, c_{i,\alpha}^\dagger c_{j,\alpha})$ where a fermion hops across a black link from site $j$ to $i$ and then back to $j$. The projection operators $(1 - \eta_{\hat{i}})(1 - \eta_{\hat{j}})$ result due to the $\eta_i \eta_{\hat{i}} = \eta_j \eta_{\hat{j}} = 0$ constraint, which prevents the virtual process from occurring when $\hat{i}$ or $\hat{j}$ is occupied by a boson. At a mean-field level, we can think of the projection operators as effectively weakening the repulsive interaction $(\eta_i - \frac{1}{2})(\eta_j - \frac{1}{2})$ and shifting the boson chemical potential $\mu'$.

Given its similarity to $H_\square^{\text{eff}}$ [Eq. (8)] in the previous section, $H_\triangle^{\text{eff}}$ is likely to also have a superfluid ground state for certain $\mu'$. However, this will have to be checked numerically, which could be done using sign-free[11] quantum Monte Carlo [89]. This implies that the original model $H_\triangle$ [Eq. (20)] is also likely to have a superconducting ground state in the limit considered in Eq. (21) for some range of $\mu'$.

## B   Effective Hamiltonian

In this appendix, we will use Schrieffer-Wolff degenerate perturbation theory [64,65] to derive the effective Hamiltonians $H_\square^{\text{eff}}$ [Eq. (8)] and $H_\triangle^{\text{eff}}$ [Eq. (22)].

The input to degenerate perturbation theory is a Hamiltonian, which is the sum of a degenerate Hamiltonian $H_0$ and a small perturbation $H_1$ to split the degeneracy:

$$H = H_0 + H_1 \tag{24}$$

---

[11]$H^{\text{eff}}$ does not have a sign problem since $-H^{\text{eff}}$, which appears in the Boltzmann factor $e^{-\beta H_\triangle^{\text{eff}}}$, has positive off-diagonal elements when viewed as a matrix in the boson number basis.

A unitary transformation can be perturbatively derived to rotate $H$ into an effective Hamiltonian that acts only on the degenerate ground state space of $H_0$:

$$H^{\text{eff}} = E_0 + \mathcal{P}H_1\mathcal{P} + \mathcal{P}H_1\mathcal{D}H_1\mathcal{P} + \cdots, \tag{25}$$

$E_0$ is the ground state energy of $H_0$; $\mathcal{P}$ projects onto the degenerate ground states of $H_0$; and

$$\mathcal{D} = \frac{1-\mathcal{P}}{E_0 - H_0} \tag{26}$$

projects into the excited states, but with an energy penalty in the denominator. A more thorough review of degenerate perturbation theory can be found in Appendix B of Ref. [90].

## B.1 Minimal Model

Here, we derive $H^{\text{eff}}_\square$ in Eq. (8).

For arbitrary $s$, $U$, and $\mu$, the four lowest energy eigenstates of Eq. (3) become

$$
\begin{array}{c|c}
|\psi\rangle & E \\
\hline
\begin{aligned}
|\tilde{0}\rangle &= \tfrac{1}{\sqrt{2}}\big(c_3^\dagger + c_4^\dagger\big)|0\rangle \\
|\tilde{1}\rangle &= \tfrac{1}{\sqrt{1+\epsilon^2}}\, c_1^\dagger c_2^\dagger(c_4^\dagger + \epsilon\, c_3^\dagger)|0\rangle
\end{aligned}
&
\begin{aligned}
-\mu - s &= -\mu' - \big(\tfrac{3}{2}-\epsilon\big)s \\
-3\mu - \epsilon s &= -3\mu' - \big(\tfrac{3}{2}-\epsilon\big)s
\end{aligned} \\
\hline
\begin{aligned}
\tfrac{1}{\sqrt{1+\epsilon_2^2}}\, c_1^\dagger(c_4^\dagger + \epsilon_2\, c_3^\dagger)|0\rangle \\
\tfrac{1}{\sqrt{1+\epsilon_2^2}}\, c_2^\dagger(c_4^\dagger + \epsilon_2\, c_3^\dagger)|0\rangle
\end{aligned}
&
\begin{aligned}
-2\mu - \epsilon_2 s &= -2\mu' + (1 - \epsilon - \epsilon_2)s
\end{aligned}
\end{array}
\tag{27}
$$

where the last two states are degenerate and we have defined

$$
\begin{aligned}
\mu' &= \mu - \frac{1-\epsilon}{2}s, \\
\epsilon &= \sqrt{1 + \left(\frac{U}{s}\right)^2} - \frac{U}{s} = \frac{s}{2U} + O\left(\frac{s}{U}\right)^3, \\
\epsilon_2 &= \sqrt{1 + \left(\frac{U}{2s}\right)^2} - \frac{U}{2s} = \frac{s}{U} + O\left(\frac{s}{U}\right)^3.
\end{aligned}
\tag{28}
$$

When $\mu' = 0$, the first two states in Eq. (27) are degenerate.

To apply degenerate perturbation theory, we define

$$H_0 = \sum_i H_i^{(4)} - H_{\mu'}, \qquad H_1 = H'_\square + H_{\mu'}, \qquad H_{\mu'} = -\mu' \sum_i \sum_{\alpha=1,2,3,4} n_{i,\alpha}, \tag{29}$$

where $H_i^{(4)}$ and $H'_\square$ are defined in Eq. (6). $H_{\mu'}$ is subtracted in the definition of $H_0$ so that $H_0$ is degenerate.

It is useful to define hard-core boson annihilation and number operators that act on the unperturbed ground states [Eq. (27)] as follows:

$$
\begin{aligned}
b_i^\dagger|\tilde{0}_i\rangle &= |\tilde{1}_i\rangle, & \eta_i|\tilde{0}_i\rangle &= 0, \\
b_i|\tilde{1}_i\rangle &= |\tilde{0}_i\rangle, & \eta_i|\tilde{1}_i\rangle &= |\tilde{1}_i\rangle, \\
b_i|\tilde{0}_i\rangle = b_i^\dagger|\tilde{1}_i\rangle &= 0, & \eta_i &= b_i^\dagger b_i.
\end{aligned}
\tag{30}
$$

The boson operators can be written in terms of the fermions as

$$b_i = c_{i,1}c_{i,2}\frac{1}{\sqrt{2}}(c^{\dagger}_{i,3} + c^{\dagger}_{i,4})(1 - n_{i,3})c_{i,4} + O\left(\frac{s}{U}\right),$$

$$\eta_i = b^{\dagger}_i b_i = n_{i,1}n_{n,2}(1 - n_{i,3})n_{i,4} + O\left(\frac{s}{U}\right). \tag{31}$$

Within the ground state space of $H_0$, into which $\mathcal{P}$ projects, the above can be simplified to

$$\mathcal{P}b_i\mathcal{P} = \sqrt{2}\,\mathcal{P}c_{i,1}c_{i,2}\mathcal{P} + O\left(\frac{s}{U}\right),$$

$$\mathcal{P}\eta_i\mathcal{P} = \mathcal{P}n_{i,1}n_{n,2}\mathcal{P}. \tag{32}$$

The $\sqrt{2}$ appears in order to cancel the $\frac{1}{\sqrt{2}}$ in $|\tilde{0}\rangle$.

The first-order correction to $H^{\text{eff}}$ [Eq. (25)] is

$$\mathcal{P}H_1\mathcal{P} = \mathcal{P}H_{\mu'}\mathcal{P} = -\mu'\sum_i\mathcal{P}\left(n_{i,1} + n_{i,2}\right)\mathcal{P} + \text{const} = -2\mu'\sum_i\mathcal{P}\eta_i\mathcal{P} + \text{const}. \tag{33}$$

Eq. (33) results because the grounds states of $H_0$ always have $\eta_i = n_{i,1} = n_{i,2}$, We also ignore the constant term since this just shifts the energies.

The next term is given by

$$\mathcal{P}H_1\mathcal{D}H_1\mathcal{P} = +t^2\sum_{\langle ij\rangle}\sum_{\alpha,\beta=1,2}\mathcal{P}\left(c^{\dagger}_{i,\alpha}c_{j,\alpha} + c^{\dagger}_{j,\alpha}c_{i,\alpha}\right)\mathcal{D}\left(c^{\dagger}_{i,\beta}c_{j,\beta} + c^{\dagger}_{j,\beta}c_{i,\beta}\right)\mathcal{P} \tag{34}$$

$$= +t^2\sum_{\langle ij\rangle}\sum_{\alpha}\left\{|\tilde{1}_i\tilde{0}_j\rangle\langle\tilde{1}_i\tilde{0}_j|c^{\dagger}_{i,\alpha}c_{j,\alpha}\frac{1-\mathcal{P}}{-s}c^{\dagger}_{i,\bar{\alpha}}c_{j,\bar{\alpha}}|\tilde{0}_i\tilde{1}_j\rangle\langle\tilde{0}_i\tilde{1}_j| \tag{35}\right.$$

$$\left. + |\tilde{1}_i\tilde{0}_j\rangle\langle\tilde{1}_i\tilde{0}_j|c^{\dagger}_{i,\alpha}c_{j,\alpha}\frac{1-\mathcal{P}}{-s}c^{\dagger}_{j,\alpha}c_{i,\alpha}|\tilde{1}_i\tilde{0}_j\rangle\langle\tilde{1}_i\tilde{0}_j| + (i\leftrightarrow j)\right\}$$

$$= +t^2\sum_{\langle ij\rangle}\sum_{\alpha}\left\{|\tilde{1}_i\tilde{0}_j\rangle\frac{1}{-2s}\langle\tilde{0}_i\tilde{1}_j| + |\tilde{1}_i\tilde{0}_j\rangle\frac{1}{-2s}\langle\tilde{1}_i\tilde{0}_j| + (i\leftrightarrow j)\right\} \tag{36}$$

$$= -\frac{t^2}{s}\sum_{\langle ij\rangle}\mathcal{P}\left[b^{\dagger}_ib_j + b^{\dagger}_jb_i - 2\left(\eta_i - \tfrac{1}{2}\right)\left(\eta_j - \tfrac{1}{2}\right) + \tfrac{1}{2}\right]\mathcal{P}. \tag{37}$$

In the above, we are ignoring terms much smaller than $O(t^2/s)$. $|\tilde{1}_i\tilde{0}_j\rangle\langle\tilde{1}_i\tilde{0}_j|$ projects the unit cell $i$ into the state $|\tilde{1}\rangle$ and $j$ into the state $|\tilde{0}\rangle$. $(i\leftrightarrow j)$ denotes a copy of the expression to its left with $i$ and $j$ interchanged. $\bar{\alpha} = 1$ when $\alpha = 2$ and $\bar{\alpha} = 2$ when $\alpha = 1$. Eq. (36) is obtained by calculating an inner product in an 8-fermion Hilbert space. Eq. (37) makes use of the ground state projection operator $\mathcal{P}$ and hard-core boson operators [Eq. (30)]. The sum over $\alpha$ just results in a factor of two.

Adding together Eqs. (33) and (37) reproduces $H^{\text{eff}}_{\square}$ in Eq. (8) up to constant terms, which we ignore in the main text.

## B.2 Triangular Model

To derive $H^{\text{eff}}_{\triangle}$ in Eq. (22), we define

$$H_0 = \sum_{\langle ij\rangle'}H^{(6)}_{ij} - H_{\mu'}, \qquad H_1 = H'_{\triangle} + H_{\mu'}, \qquad H_{\mu'} = -\mu'\sum_i\sum_{\alpha=1,2,3}n_{i,\alpha}, \tag{38}$$

where $H^{(6)}_{ij}$ and $H'_{\triangle}$ are defined in Eq. (20) and $\mu' = \mu - s/2 + O(s^2/U)$. We will work using the limit in Eq. (21), and derive $H^{\text{eff}}_{\triangle}$ up to corrections of order $O(t^4/s^3)$ and $O(s^2/U)$.

We will define hard-core boson annihilation and number operators that act on the unperturbed ground states [Eq. (19)] as follows:

$$
\begin{aligned}
b_j^\dagger |\widetilde{00}_j\rangle &= |\widetilde{01}_j\rangle, & \eta_j |\widetilde{00}_j\rangle &= 0, \\
b_j |\widetilde{01}_j\rangle &= |\widetilde{00}_j\rangle, & \eta_j |\widetilde{01}_j\rangle &= |\widetilde{01}_j\rangle, \\
b_j^\dagger |\widetilde{01}_j\rangle &= b_j^\dagger |\widetilde{10}_j\rangle = 0, & \eta_j |\widetilde{10}_j\rangle &= 0, \\
b_j |\widetilde{00}_j\rangle &= b_j |\widetilde{10}_j\rangle = 0, & \eta_j &= b_j^\dagger b_j.
\end{aligned}
\tag{39}
$$

If $\hat{j}$ is the 3-site cluster across a red link [shown in Eq. (20)] from $j$, then $b_{\hat{j}}$ acts similarly but on the first digit in the ket; e.g. $b_{\hat{j}}^\dagger |\widetilde{00}_j\rangle = |\widetilde{10}_j\rangle$. Note that within the above Hilbert space, the following constraint is obeyed: $\eta_{\hat{j}} \eta_j = 0$. The boson operators can be written in terms of the fermions as

$$
\begin{aligned}
b_j &= c_{j,1} c_{j,2} \tfrac{1}{\sqrt{2}} (c_{j,3}^\dagger + c_{\hat{j},3}^\dagger)(1 - n_{\hat{j},3}) c_{j,3}, \\
\eta_j &= b_j^\dagger b_j = n_{j,1} n_{j,2} (1 - n_{\hat{j},3}) n_{j,3}.
\end{aligned}
\tag{40}
$$

Within the ground state space of $H_0$, into which $\mathcal{P}$ projects, the above can be simplified to

$$
\begin{aligned}
\mathcal{P} b_j \mathcal{P} &= \sqrt{2}\, \mathcal{P} c_{j,1} c_{j,2} \mathcal{P}, \\
\mathcal{P} \eta_j \mathcal{P} &= \mathcal{P} n_{j,1} n_{j,2} \mathcal{P}.
\end{aligned}
\tag{41}
$$

The first non-constant term of $H^{\text{eff}}$ in Eq. (25) is

$$
\mathcal{P} H_1 \mathcal{P} = -\mu' \sum_i \mathcal{P} (n_{i,1} + n_{i,2}) \mathcal{P} = -2\mu' \sum_i \mathcal{P} \eta_i \mathcal{P}.
\tag{42}
$$

Eq. (42) results because the grounds states of $H_0$ always have $\eta_i = n_{i,1} = n_{i,2}$.

The next term is given by

$$
\mathcal{P} H_1 \mathcal{D} H_1 \mathcal{P} = +t^2 \sum_{\langle ij \rangle} \sum_{\alpha,\beta=1,2} \mathcal{P} (c_{i,\alpha}^\dagger c_{j,\alpha} + c_{j,\alpha}^\dagger c_{i,\alpha}) \mathcal{D} (c_{i,\beta}^\dagger c_{j,\beta} + c_{j,\beta}^\dagger c_{i,\beta}) \mathcal{P}
\tag{43}
$$

$$
= +t^2 \sum_{\langle ij \rangle''} \sum_\alpha \Big\{ |\widetilde{01}_i \widetilde{00}_j\rangle \langle \widetilde{01}_i \widetilde{00}_j | c_{i,\alpha}^\dagger c_{j,\alpha} \frac{1-\mathcal{P}}{-s} c_{i,\bar{\alpha}}^\dagger c_{j,\bar{\alpha}} |\widetilde{00}_i \widetilde{01}_j\rangle \langle \widetilde{00}_i \widetilde{01}_j |
\tag{44}
$$

$$
+ |\widetilde{01}_i \widetilde{00}_j\rangle \langle \widetilde{01}_i \widetilde{00}_j | c_{i,\alpha}^\dagger c_{j,\alpha} \frac{1-\mathcal{P}}{-s} c_{j,\alpha}^\dagger c_{i,\alpha} |\widetilde{01}_i \widetilde{00}_j\rangle \langle \widetilde{01}_i \widetilde{00}_j | + (i \leftrightarrow j) \Big\}
$$

$$
= +t^2 \sum_{\langle ij \rangle''} \sum_\alpha \Big\{ |\widetilde{01}_i \widetilde{00}_j\rangle \frac{1}{-2s} \langle \widetilde{00}_i \widetilde{01}_j | + |\widetilde{01}_i \widetilde{00}_j\rangle \frac{1}{-2s} \langle \widetilde{01}_i \widetilde{00}_j | + (i \leftrightarrow j) \Big\}
\tag{45}
$$

$$
= -\frac{t^2}{s} \sum_{\langle ij \rangle''} \mathcal{P} \Big[ b_i^\dagger b_j + b_j^\dagger b_i - 2(1-\eta_{\hat{i}}) \underbrace{(\eta_i \eta_j - \tfrac{1}{2}\eta_i - \tfrac{1}{2}\eta_j)}_{(\eta_i - \frac{1}{2})(\eta_j - \frac{1}{2}) - \frac{1}{4}} (1-\eta_{\hat{j}}) \Big] \mathcal{P}.
\tag{46}
$$

We are neglecting small $O(U^{-1})$ terms. In Eq. (44), $\sum_{\langle ij \rangle'}$ (and $\sum_{\langle ij \rangle''}$) sum over the neighboring 3-site clusters with (and without) a red line between them in Eq. (20). $|\widetilde{01}_i \widetilde{00}_j\rangle \langle \widetilde{01}_i \widetilde{00}_j |$ projects the 6-site unit cell $(\hat{i}, i)$ into the state $|\widetilde{01}\rangle$ and $(\hat{j}, j)$ into the state $|\widetilde{00}\rangle$ [Eq. (19)]. $\hat{i}$ denotes the 3-site cluster across a red link from $i$ in Eq. (20), and similar for $\hat{j}$ and $j$.

Adding together Eqs. (42) and (46) reproduces $H_\triangle^{\text{eff}}$ in Eq. (22) up to constant terms, which we ignore in Appendix A.

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
