# Peer review of "A Simple Mechanism for Unconventional Superconductivity in a Repulsive Fermion Model"

_SciPost Physics, doi:SciPost Phys. 6, 016 (2019)_

## Round 4 · Referee Report · Anonymous (Referee 1) · 2018-10-23

Strengths

1) The manuscript produces a simple model to understand superconductivity from repulsive interaction. 2) Overall well-written and well-organized work, which is easy to follow. Building up a case for superconductivity from simple local clusters to lattices and then extending with some perturbations.

Weaknesses

1) The manuscript starts with a simple 4-site model with a degenerate ground state acting as a hard-core boson state with 0 or 1 bosons, respectively. However, this is only achieved at the extreme fine-tuning point of mu = s. Then, coupling such 4-site models into a lattice the authors again reach a hard-core model to order t^2/s (Eq. 7) at the fine-tuning point mu = s/2. The authors then try to improve on the model to make it more general. However, to me a large limitation seems to be this extreme fine-tuning. What happens if mu is not exactly s or s/2? Are the results approximately valid or is it crucial to have an exact ground state degeneracy? 2) The manuscript derives effective attractive Hamiltonians for hard-core bosons and then use previous results to proclaim a superfluid state which is argued to be a superconductor as well. It would be good if the manuscript is more explicit on this in order to clarify the superconducting /superfluid state. For example on p. 1 it is mentioned that a bosonic bound state is formed, which then condenses. However, on p. 2 it seems to be stated that the condensate has p_z symmetry, which would then mean overall Fermi-Dirac statistics for the condensate (since it’s spinless)? In short, will the condensate be bosonic or fermionic and what are the implications for the order parameter symmetry? 3) How unimportant are the ignored terms in the effective Hamiltonians? For example in Eq. 4 terms of order s^2/U are ignored and similarly in Eq. 7. In fact, from the derivation of Eq. 7 in the Appendix it is not even obvious that higher order terms are smaller. Can this also complicate the fine-tuning that is need (see point 1)? 4) In sections 4.1-4.3 perturbations are added to the system, but how do these effect the exact degeneracy that is needed for the ground state in the very first step of the argument (see also point 1)? For example in Eq. 8, why not solve the 4-site model exactly and show that the ground state is still exactly degenerate? 5) A large limitation to achieve anything similar to this model in real materials seems to be the spinless nature of the pairing state. Section 4.3 is spent trying to remedy this, but beyond looking at only a single cluster, the results seems to only be speculations? 6) One of the few actual materials mentioned for this work is doped C60. Are these the fulleride crystals? It would be very interesting if the authors can elaborate a bit more on the said similarities between this work and superconductivity in the fullerides. For example, how can the fullerides be considered to be spinless superconductors?

Report

The manuscript provides an interesting idea to generate superconductivity form repulsive interactions by explicitly showing how a clever cluster can generate an attractive model for hard-core bosons. Extending this to lattices requires for some necessary approximations and one weakness is that it is not clear how accurate this result is then (see e.g. points 1,3 under weaknesses). Also, while the authors brings up a set of possible perturbations and show that the results still hold, it is still very unclear how feasible it is to even try to apply this model framwork to any sort of material. For example the spinless property is problematic, but so is the intricate orbital/site structure needed. In fact, many superconductors with multiple low-energy orbitals have an intricate dependence, even for the order parameter, on the orbital structure, for example producing odd-parity interorbital pairing states in Cu-doped Bi2Se3. The competition with such states are seemingly not taken into account here.

Requested changes

1) In the introduction there is no mentioning of mechanisms causing pairing from extremely strong repulsive interactions, such as the t-J model. For completeness such references would be appropriate. 2) Below Eq. 3, it is never explained why this is a p_z Cooper pair, and in what direction is z? 3) Why call the state nodeless and not just gapped? 4) Several figure texts are missing. 5) The logic leading up to Eq. 7 seems to be reversed: saying H’ must act twice and thus perturbation theory to order t^2/s is enough. I believe the final result might be ok, but the argument is strange. 6) Why is n_5 missing in the last line in Eq. 10? 7) In Section 5 it is first stated that Tc~t/2, but in the footnote it says Tc~t, which one is correct?

---

## Round 4 · Referee Report · Anonymous (Referee 2) · 2018-11-6

Strengths

The authors introduce a tight-binding model of fermions with repulsive interactions that exhibits unconventional SC.

(1) A main result is that the model produces pseudo-gap-like physics above the transition temperature, with fermions remaining paired and with a large single-particle energy gap.

(2) The main model—a 4 site model of spinless fermions, is introduced in Eq.(2). The discussion that follows shows that this model has features that mimic superconductivity (at least in terms of energetics), and in a certain limit produces and effective attractive interaction.

Weaknesses

(1) The model is not physical motivated, i.e., the sort of miscroscopic crystal and local orbital structures that would lead to this are not clearly motivated. It seems the motivation is simply to produce a model with the desired features. Once this is established, the authors go on to study a grid of coupled 4-site models form a two-dimensional square lattice.

Section 4 of the paper goes on to address the lack of realism of the model by discussion some extensions of it.
(2) Mostly this section uses order of magnitude estimates to argue what the behavior of the system will be in various parameter limits. Very few numerical calculations or numerically generated phase diagrams are exhibited.

(3) I also found missing a detailed discussion of how this work relates to prior work on repulsive SC, such as in Ref [25-27].

Report

In summary, I found the paper did not present a clear message of what this model accomplishes and where it falls short. It would also be helpful to better relate this model to other repulsive fermion models of SC for comparison/contrast. How is this model “better” than prior models of repulsive Hamiltonians supporting SC?

Requested changes

I recommend the authors address the weaknesses before publication.

---

## Round 5 · Referee Report · Anonymous · 2019-1-15

Report

In my view, the authors have substantially improved the manuscript in response to my comments and those of the other referee. It is much better motivated, easier to read, and more complete in its description of the model assumptions, main results, and contact to physical systems. I do not have any further suggestions for improvement.

---

## Round 5 · Referee Report · Anonymous · 2019-1-24

Report

I am pleased to see that the authors have significantly improved on their manuscript, as well as provided well thought through answers to most of my previous listed concerns.

I still find the model very contrived though, although I understand the authors goal was to provide a simple model that still produces superconductivity from strong repulsion.

In particular, I am now confused with the additional motivation that this model might be describing the physics in a conductor-dielectric-conductor trilayer. If that is the case, why is only site 3 in the dielectric coupling to the conductors and not site 4? It seems as this asymmetry between sites 3 and 4 is essential, still there is seemingly no motivation to how this can be achieved, neither in the text nor in Fig. 1. Is perhaps the 5-site cluster in section 4.2 addressing this issue? However, I’m not fully following the logic of this section, for example it says that site 3 can be thought of as the superposition of px orbitals of ions 3 and 4. But, what motivates this asymmetry between sites 4 and 5 and why is again not sites 4 and 5 coupling to sites 1 and 2? A thorough explanation to this would be very helpful.

  • validity: high
  • significance: good
  • originality: high
  • clarity: good
  • formatting: excellent
  • grammar: perfect

Author:  Kevin Slagle  on 2019-01-25  [id 413]

(in reply to Report 2 on 2019-01-24)
Category:
answer to question
correction

The asymmetry in the 4-site model is important and can result if site 4 is physically further from sites 1 and 2 than site 3 is, which would reduce the hopping and Hubbard repulsion between site 4 and 1 (and between 4 and 2). The 5-site cluster is indeed more symmetric, and the same logic can be applied.

We apologize for a typo which may have resulted in confusion. We meant to say that "site 3 can be thought of as the superposition of px orbitals of ions 4 and 5" (rather than ions 3 and 4). We will fix this typo in a minor revision.

---

## Round 5 · Author Response

Dear Editor and Referees,

Thank you for carefully reviewing our work and for the detailed comments and suggestions. We have taken the revision requests very seriously and we are submitting a new draft with major revisions. We believe that the new draft establishes a clear theoretical and physical motivation and that we have fully addressed the concerns of the referees in the responses below.

Best regards, Kevin Slagle and Yong Baek Kim

Response to Referee 1:

Weakness 1: Physically, there is no fine tuning. The physics is robust to all local perturbations. The exact degeneracy in the 4-site model was just a trick so that we could do degenerate perturbation theory. We now emphasize and explain this just before Section 4. If mu is not exactly equal to s/2, then this just shifts the chemical potential mu' in the hard-core boson model, which is more explicit in the new draft.

Weakness 2 and requested changes 2 and 3: We originally stated that the Cooper pair can be thought of as being nodelss p_z-wave since there is antisymmetry in the layer index (in analogy to the nodeless d-wave pairing in Ref. 33 (Ref. 37 in the new draft)). (The sites 1 and 2 in the 4-site model are displaced in the z-direction.) However, after talking to more of the community, it seems that this may be misleading. We now refer to the Cooper pair as s-wave, where the antisymmetry is in the alpha=1,2 index. The Cooper pair is a boson, and the condensate is a bosonic condensate as usual.

Weakness 3: We mistakenly previously derived the effective boson model while also assuming s/U << t/s. In the new draft, we drop this rather excessive assumption. Dropping this assumption just adds the epsilon to the definition of mu' in Equation 8 of the new draft. Now, all ignored terms are much smaller than O(t^2/s) and are safe to ignore.

Weakness 4: In all three of these subsections, we explain how to obtain the exact degeneracy. Theoretically, obtaining a degeneracy is easy by just tuning the chemical potential. The hard part is making sure that the degenerate states have a difference in fermion number equal to two and a large pair-binding energy (see Figure 2b). The solutions are exact in the limits considered in each section. See also our response to weakness 2 of the other Referee.

Weakness 5: The spinless nature of the pairing state in our model is actually an advantage since the unimportance of spin could allow a superconducting realization of our model to be more robust to magnetic fields. As mentioned at the beginning of Section 4.3, spinless fermions could be effectively obtained by applying an in-plane magnetic field. However, Section 4.3 shows that a hare-core boson model (where the bosons have 4 spin states) can also be obtained in a similar way using spinful fermions. It is very likely that this hard-core boson model also hosts superconductivity. The new draft makes this claim more explicit, and also includes a more detailed example in Section 4.3.1 that reproduces the hard-core boson found in Section 3, which is known to exhibit superconductivity.

Weakness 6: Yes, we are making reference to the fullerene crystals. The similarity to the 4-site model is not related to spinless fermions, but the fact that both models (with the right chemical potential) consist of clusters with low-energy states that have a difference in fermion number equal to two.

Report: We hope the referee now finds that our results are accurate and that a conductor-dielectric-conductor trilayer may be a feasible realization of our toy model.

Requested change 1: We added two paragraphs to the introduction that discuss the t-J and Hubbard models and help motivate our model.

Requested change 2 and 3: See weakness 2.

Requested change 4: All of the figures have captions. Many of the equations have small graphics that accompany them. We have added a little extra clarification below Equation 6 in the new draft.

Requested change 5: We reworded that paragraph to be more explicit. We just meant that the leading order process resulted in a term of order t^2/s.

Requested change 6: Thank you bringing this typo to our attention. It is fixed now.

Requested change 7: Tc~t/2 is more precise. We corrected the footnote.

Response to Referee 2:

Weakness 1: Indeed, the original motivation of this work was primarily theoretical: to uncover the simplest analytically-tractable theoretical model of unconventional superconductivity. In the new draft, we added additional physical motivation for a conductor-dielectric-conductor trilayer. See Figure 1 and the second paragraph of Section 5.2.

Weakness 2: Perhaps our motivation for this section was not clear. In Section 4, our claim is that those extensions to our model also result in hard-core boson models in the strong interaction limit, and that the hard-core boson models are known to or are very likely to superconduct. We have provided sufficient analysis to support this claim. We have made our motivation for this section more clear in the new draft.

Weakness 3: In the new draft, we have added additional discussion to the introduction to discuss this relation.

Report: The model is significantly simpler and more analytically tractable than previous strongly interacting models of superconductivity. However, it is just a toy model, and future work is needed to explore the possibility of a conductor-dielectric-conductor trilayer realization. We added comparisons to other models in the introduction.

---

## Round 5 · List of Changes

1) The introduction has been expanded to discuss previous models of unconventional superconductivity in order to motivate our model.

2) We now physically motivate our model as a toy model for a conductor-dielectric-conductor trilayer in the abstract, introduction, Figure 1, after Equations 2 and 6, and the Discussion (Section 5.2).

3) In response to Referee 1 and private communications, we have clarified the nature of the superconducting order parameter below Equation 3.

4) Figure 2 and Equation 5 have been added to help emphasize the generality of the 4-site model to smaller repulsion.

5) A comment regarding fine tuning was added at the end of Section 3.

6) We further emphasized our motivations in the abstract and Sections 4 and 5.

7) Section 4.3.1 was added.

8) In response to weakness 3 that Referee 1 pointed out, the perturbation theory in he appendix has been slightly reformulated.

A detailed markup of changes to the text can be found at goo.gl/5Wg2bE

---

## Editorial Decision

published